# Mobility evaluation by GPS tracking in a rural, low-income population in Cambodia

**Anaïs Pepey**[1]*, **Thomas Obadia**[2,3], **Saorin Kim**[1], **Siv Sovannaroth**[4], **Ivo Mueller**[5,6], **Benoit Witkowski**[1], **Amélie Vantaux**[1], **Marc Souris**[7]

**1** Malaria Molecular Epidemiology Unit, Institut Pasteur du Cambodge, Phnom Penh, Cambodia, **2** Department of Parasites and Insect Vectors, Infectious Diseases Epidemiology and Analytics, Institut Pasteur, Paris, France, **3** Département de Biologie Computationnelle, Hub de Bioinformatique et Biostatistique, Institut Pasteur, Paris, France, **4** National Centre for Parasitology Entomology and Malaria Control (CNM), Phnom Penh, Cambodia, **5** The Walter and Eliza Hall Institute of Medical Research, Parkville, Victoria, Australia, **6** Department of Medical Biology, University of Melbourne, Melbourne, Victoria, Australia, **7** UMR Unité des Virus Emergents, UVE: Aix-Marseille Univ–IRD 190–Inserm 1207–IHU 5 Méditerranée Infection, Marseille, France

* apepey@posteo.net

**Data Availability Statement:** The datasets are available on figshare at the following DOI: https://doi.org/10.6084/m9.figshare.19538020.v1.

## Abstract

Global Positioning System (GPS) technology is an effective tool for quantifying individuals' mobility patterns and can be used to understand their influence on infectious disease transmission. In Cambodia, mobility measurements have been limited to questionnaires, which are of limited efficacy in rural environments. In this study, we used GPS tracking to measure the daily mobility of Cambodian forest goers, a population at high risk of malaria, and developed a workflow adapted to local constraints to produce an optimal dataset representative of the participants' mobility. We provide a detailed assessment of the GPS tracking and analysis of the data, and highlight the associated difficulties to facilitate the implementation of similar studies in the future.

## Introduction

The study of individuals' daily movement patterns is important for understanding the temporal and spatial scales of infectious disease transmission [1,2]. The use of Global Positioning System (GPS) technology to assess these movement patterns has become increasingly common in infectious disease research projects due to its accessibility and convenience, for example in the context of various vector-borne transmission studies for dengue [3,4] and malaria [5–7]. The validity of various models has been assessed under different environmental conditions [8]. These devices offer various advantages compared to other methodologies, like cell phones, maps or questionnaires. Indeed, GPS data loggers are not subject to recall bias and do not depend on spatial literacy in the study population or on the existence of addresses and maps in the study area, which can be limited in resource-poor and low-income populations [9]. Compared to GPS devices, cell phones are another option but are more expensive and more prone to theft. Relying on personal cellular devices might furthermore result in biased sampling of the population [10]. In addition, the encryption of cell phone positional data is harder to

**Funding:** This study was funded by the National Institutes of Health program International Centres of Excellence for Malaria Research, grant 1U19AI129392-01 "Understanding, tracking and eliminating malaria transmission in the Asia-Pacific Region". A.P. was funded by the Pasteur Institute International Network Calmette and Yersin Ph.D. scholarship, grant MHL/MJ/N°212/18. The funders had no role in study design, data collection and analysis, decision to publish, or preparation of the manuscript.

**Competing interests:** The authors have declared that no competing interests exist.

institute [3] and the mobile network in remote or rural areas is often partial [11], which would deteriorate spatial data recording [10].

Though GPS tracking is an effective methodology for the collection of spatial data for vector-borne disease research, it also comes with limitations: timing and orbital errors, inadequate satellite geometry and signal inconsistencies can generate bias or errors [8,12]. Human-induced error is also to be expected, as participants could forget to carry the device.

Some recommendations already exist for handling GPS devices and data processing, but do not include the difficulties of implementing the devices in the study population [13]. A previous study described a GPS follow-up implementation to compare their efficacy to semi-structured interviews in a population sampled in urban South America [4]. The best practices to implement GPS tracking to follow forest goers, in accordance to their working conditions and to a rural environment, are not yet available in the literature.

While Cambodia aims to eliminate malaria by 2025, several pockets of transmission remain in the country, with the highest incidence rate found in Mondulkiri, Eastern Cambodia [14]. Forest goers—represented by young and adult males [15–17]—have repeatedly been shown to be the most at-risk population for malaria in Cambodia [15,18–21]. Malaria vectors (*Anopheles spp.*) are found mostly in forest environments [22]: forest activities are therefore the most important risk factor. Malaria vectors also inhabit forest fringes [22], putting individuals in farmer huts and engaging in plantation work at risk of exposure to infectious mosquito bites [23]. *Anopheles* vectors are mostly nocturnal species, but they can also be active during dusk and dawn, outside the hours of bed net use [24,25]. Malaria risk exposure can vary from one village to another [16,26] but also between households [27], highlighting the need for more precise tools to measure such local heterogeneity. Individuals' traversal of a study area, in and around forest patches at different times of day, needs to be assessed and analysed to estimate individual exposure risk.

In Cambodia, the standard methodology to measure mobility has been through questionnaires [15,19–21]. To understand the role of daily movement patterns on its transmission, we implemented a GPS follow-up study to track the population at risk. We described and characterised individuals' mobility patterns, and integrated the data into a Geographic Information System (GIS) to be combined and analysed with environmental data. We furthermore documented the methodology and the limitations of the implementation to facilitate the design of future GPS tracking studies in similar rural and resource-poor environments.

## Material and methods

### Study site and sample population

The study took place in Kaev Seima district, Mondulkiri province, North-East Cambodia. Here, the climate is divided into rainy (May-October) and dry (November-April) seasons. Kaev Seima district is a rural, low-income environment where populations mainly live from agriculture and forest products exploitation. Study participants were men between the age of 13 and 60 years old to match the main malaria risk population [15,18–21]. The enrolment occurred in two rounds to characterise both seasons: first round during the rainy season (April to September 2018) with 160 participants, and second round during the dry season (February to April 2019) with 200 participants.

### Study design

We implemented GPS tracking in a rural area of Cambodia, following all participants' movements for two weeks by providing them a GPS data-logger that recorded their position. In accordance with previous guidelines [13], after the two weeks, participants were asked to fill

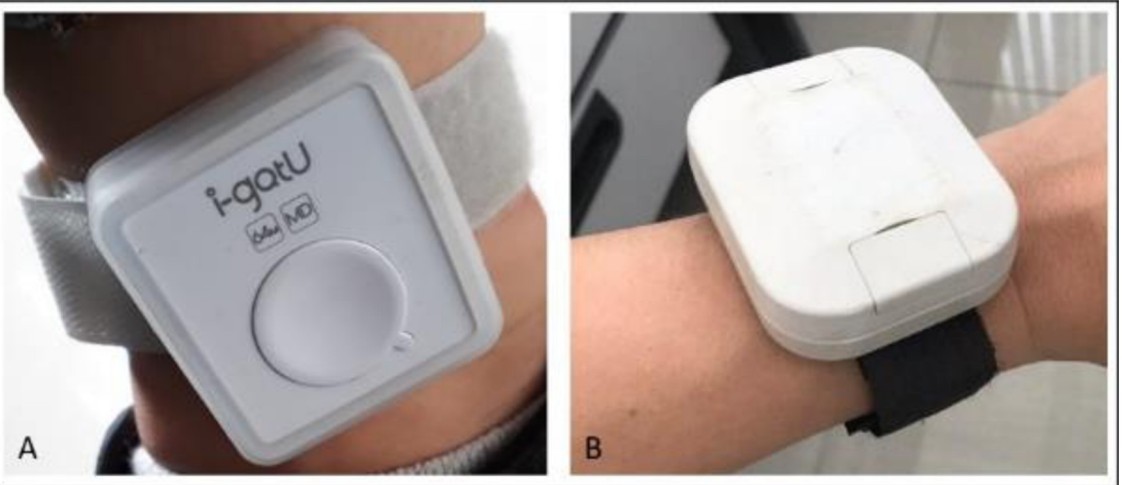

**Fig 1. Devices distributed to the study population.** A: I-gotU GT600, distributed for rainy season (here worn on an individual's ankle), B: Takachi waterproof enclosure containing the I-gotU GT600 distributed for dry season (here worn on an individual's wrist). The waterproof enclosure was not yet used during rainy season because the devices were sold as "water-resistant" and were expected to be adapted to study conditions.

out a questionnaire containing questions related to their use of GPS devices. Similar to other studies of this kind [5,6], the I-gotU GT600 (Mobile Action Technology Inc., Taipei, Taiwan, Fig 1) was used to record participants' mobility. Its battery life (up to a week), data storage (262,000 waypoints), price (< 50 USD), precision (< 10m) and unobtrusive design (37g) corresponded best to our study design. Each participant was assigned two GPS data loggers over course of the study (one for each of the two 7 day tracking periods). Serial numbers of the GPS data loggers were matched to participant unique identification numbers. The GPS loggers were password-protected, allowing only the research team to access the data. The settings intentionally prohibited the devices to be turned off, thus allowing the devices to record the mobility of the field team between households and base. Curation of the spatial data was performed with R [28].

Geographical coordinates were logged every 30 seconds. A smart tracking mode allowed for changing of logging intervals to 10 seconds when an individual's speed exceeded 10 km/h. The GPS data loggers were programmed to be motion activated and to hibernate when not in motion to better preserve battery life.

Waterproof enclosures (Takachi IP67 enclosures, reference WP5-7-2G) were used during the second round of collection to protect the loggers from precipitation and immersion (Fig 1). We fixed a Velcro strap on the plastic enclosure of the device to allow it to be worn on one's wrist or ankle.

### GPS accuracy and precision tests

To quantify the spatial exactness of the GPS data loggers, two values should be measured: accuracy and precision. Accuracy represents the closeness between the measurement and a true value, while precision describes the closeness between different measurements of a same location. Typically, GPS accuracy is determined by comparing the positional coordinates recorded by a GPS receiver to that of a known location [8]. Such location could not be found in our study area, we thus had to rely on other methods, such as comparing records from the GPS

data loggers and the most accurate GPS device available, a Garmin GPSMAP® 64s. Both precision and accuracy may depend on the type of surrounding environment.

The manufacturer did not provide any information about the device accuracy or precision. However, Duncan *et al.* (2013) tested the I-gotU GT600's stationary accuracy under various environmental conditions and found a mean error of 19.6m (SD +/- 30.9m) and a mean circular error of 10.8m, which varied from 3.3m (open sky) to 47.1m (between high-rise buildings). Vazquez-Prokopec *et al.* [3] also tested the device's spatial accuracy and obtained a stationary root mean square error of 4.4m for their stationary test and of 10.3m for their motion test along a known horizontal linear path.

To control for GPS accuracy and precision, we selected four geodic sites, one per different type of environment. In each, two to four units were left side by side at ground level to record their position for 30 minutes. The devices' hibernation mode was deactivated and their location was sampled using the Garmin GPS device. We repeated the recording process on eight occasions for each land environment category. The selected sites were: i) under a traditional Khmer house elevated three metres from the ground, where members of the households meet for social interactions; ii) in an open space next to the road in a village; iii) in a rubber plantation where trees reached three to four metres in height; and iv) in the primary forest under a sparse six-metre tall canopy.

To measure the precision, we computed the mean and standard deviation of distances to the mean centre of the recorded points. To measure the accuracy, we computed the mean distance and standard deviation to a reference point in addition to the percentage of logged points that were included in a 5-metre, 10-metre or 20-metre radius around the reference point [29]. Calculations were performed in R [28], using the distGeo function from the geosphere package [30].

## Data management

From the coordinates downloaded from the data loggers, GPS points were assembled into segments and segments into routes (or tracks). Data curation was conducted along this process, removing all points falling outside the study area or segments exceeding 150 km/h, a speed assumed not attainable within the study area and most probably arising from device recording errors (concordant with the speed threshold value recommended in [31]).

Segments were categorized based on time of capture: day (between 06h00 to 17h59) or night (between 18h00 and 05h59). Segment speeds were categorized as "slow" ($\leq$5 km/h, i.e. when participant was motionless or walking) and "fast" (when speed was >5 km/h, i.e. movement in a vehicle).

Each GPS data track was categorised depending on the logged duration (hours) and distance (kilometres). The total duration of GPS track was classified as suboptimal (<48h) or optimal ($\geq$48h) depending on the total hours logged over one week. Similarly, the average distance per day was classified as suboptimal (<10km) or optimal ($\geq$10km). Finally, only tracks categorized as optimal for both the total duration and the average distance per day were categorised as optimal. These categories allowed to dismiss the data from: *i*) devices for which the cumulative duration of records could fail to provide sufficient insights into a participant's mobility and *ii*) participants that left the device at home rather than taking it with them.

Using SavGIS [32], the time spent into each type of land use (described in Pepey et al. [33]) was calculated for each track. A 100m-resolution grid was produced to cover the study area, with each cell attributed to the major corresponding land use category (Fig 2).

We calculated for each dataset (optimal and suboptimal) the proportion of time spent: i) at slow speed versus fast speed; ii) during day time versus night time; iii) in each land use

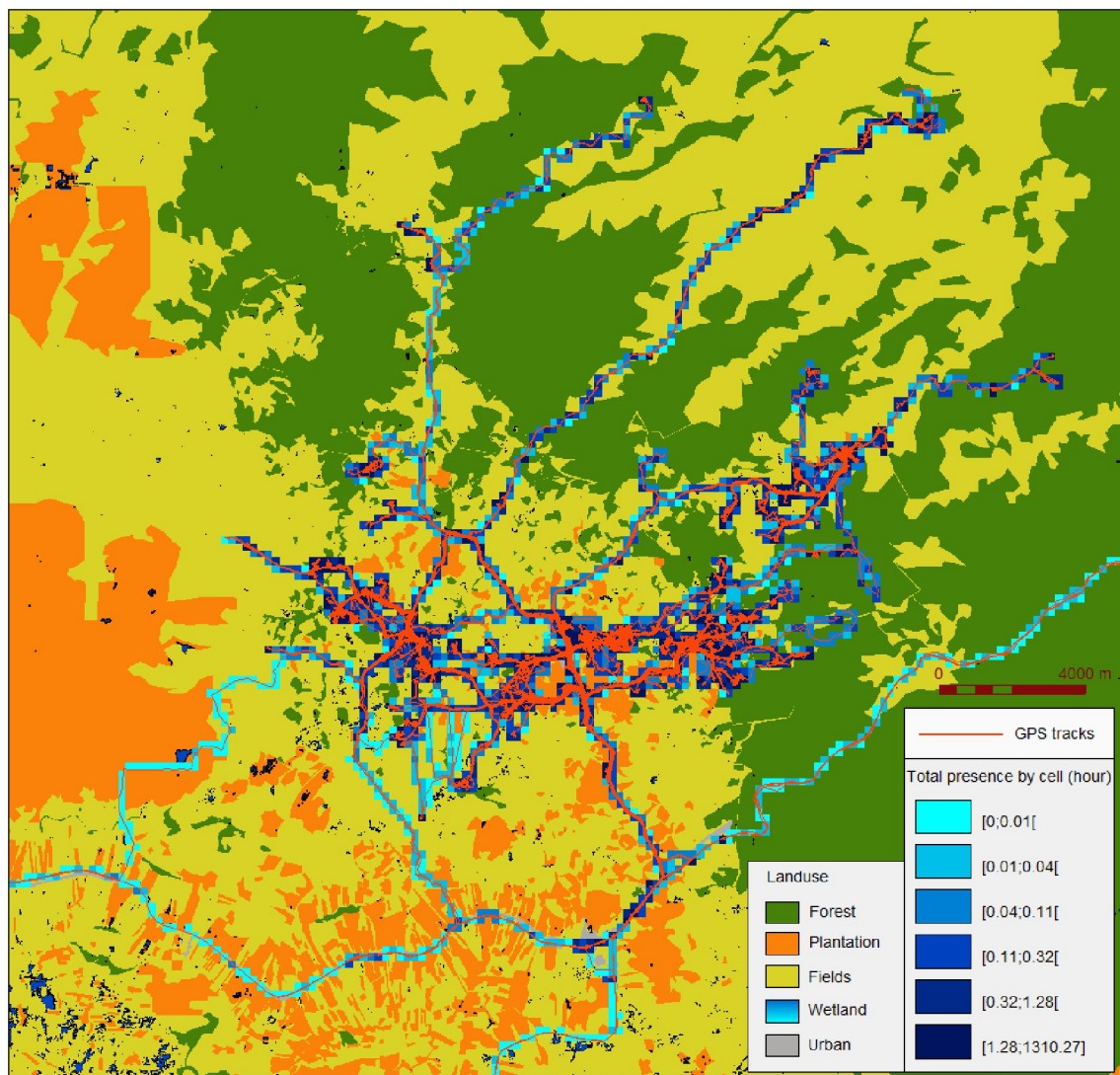

**Fig 2. GPS tracks and total presence by cell from all participants during rainy season ($N_{tracks}$ = 249, $N_{participants}$ = 151).** Land use reprinted from [33] under a CC BY license, with permission from creators of the map A. Pepey *et al*, original copyright 2020.

category. These distributions were compared using Pearson's Chi-squared test and accounting for multiple comparisons using the Bonferonni correction.

## Ethics approval and consent to participate

Ethical approvals were obtained from the Cambodian National Ethics Committee for Health Research (045NECHR & 312NHECR) and from the National Institutes of Health (DMID Protocol Number: 17–0036). The protocols conformed to the Helsinki Declaration on ethical principles for medical research involving human subjects (version 2002) and informed written consent was obtained for all volunteers. Parental assent was obtained for every participant between 13 and 18 years old.

**Table 1. Distance and duration logged by participants over the 2-week GPS follow-up.**

| Variable | Unit | Value | Rainy season | Dry season |
|---|---|---|---|---|
| Distance | km | Min | 0 | 0 |
| | | Q1 | 68.2 | 74.6 |
| | | Median | 142.1 | 127.8 |
| | | Q3 | 234.4 | 205.5 |
| | | Max | 767.1 | 488.6 |
| Time | days | Min | 0 | 0 |
| | | Q1 | 3.6 | 4.9 |
| | | Median | 6.5 | 8.4 |
| | | Q3 | 9.4 | 11 |
| | | Max | 14.8 | 16.7 |
| Distance/day | km | Min | 0 | 0 |
| | | Q1 | 16.3 | 11.8 |
| | | Median | 23.6 | 18.1 |
| | | Q3 | 32.7 | 29.4 |
| | | Max | 74.3 | 78.3 |

Durations exceeding 14 days are due to the unavailability of participants on the scheduled date, leading to postponed visits.

## Results

### Overview

Over the two seasons 4,383,582 GPS data points were logged, 59.9% of which were retained after data curation (S1 Table). Median distance logged by participants were consistent across seasons but varied greatly across participants (Table 1).

The average battery life of GPS devices was 3 day and 7 hours in the rainy season, and 4 days and 1 hour in the dry season, with great variability ranging from total malfunction (0m logged) to 16 days and 18 hours. The minimal non-null duration recorded by a GPS device was 1m and 1s, recording a total of 3 GPS points, during rainy season.

### GPS unit performance

Across all sites, the GPS devices were accurate within 20m for at least 90% of points. The stationary error of devices left in open space (accuracy: 6.1m, precision: 5.6m) or under house (accuracy: 5.9m, precision: 5m) were similar to values reported previously [3,8]. The GPS devices' logged coordinates varied between environments, with the greatest average error in plantations, for which the percentage of accurate points within a 20m radius was less than 80%. Accuracy and precision measurements for each land type can be found in S2 and S3 Tables, respectively.

### Data loss

Of the 360 enrolled participants, 261 returned both devices, allowing for a complete dataset to be downloaded (Table 2).

For the other 99 participants, an incomplete dataset was obtained. Data loss was mostly due to water infiltration and other dysfunctions from devices (77 devices, representing 69.3% of missing data, Table 3). However, data loss also arose from participants losing the device,

**Table 2. Count of participants' available GPS tracks per season and for all data.**

| Participants GPS datasets | Rainy season | Dry season | All |
|---|---|---|---|
| 2 weeks logged | 98 | 163 | 261 |
| 1 week logged | 53 | 34 | 87 |
| None | 9 | 3 | 12 |
| Total collected GPS tracks | 249 | 360 | 609 |

withdrawal or being unreachable after baseline. Eight participants withdrew because they or their families were against pursuing the study (Table 3).

In the first round of collection, 54 GPS data loggers were heavily damaged by water entering the device. This resulted in 22.2% of the data being unusable. By using waterproof enclosures to protect the GPS devices during the second round of collection, we minimised the data loss to 10%.

## Optimal and suboptimal datasets

We assigned 393 tracks (64.5%) as optimal (S4 Table). Optimal data averaged 7 days of logging per participant (range: 2 to 15 days) for a mean of 176.6 km per participants over the follow-up (25.4 to 767.1 km). Logged optimal GPS data totalised 48,399 km and 46,337 hours. The percentage of time logged in built-up areas was significantly lower for the optimal dataset, and higher for the other land use categories (S5 Table). Optimal dataset also had a significantly higher proportion of points logged during day time and above 5km/h (S5 Table).

## Questionnaire

Only 35 (9.7%) participants declared at least one removal of the device during the follow-up (21 during the rainy season and 14 during the dry season), with an average of two removals per participant per follow-up. Five participants declared that the removal was due to the discomfort, one because of its broken strap, while the 29 others did not indicate a reason. According to the number of removals and estimated time of removal given by the participants, we estimated that the data loss averaged 4 hours 36 minutes per participant per 2-week follow-up (min = 4 minutes, median = 2 hours, max = 20 hours). However, the data recorded from these participants was more complete than the global dataset: only 11.5% of them had a suboptimal length per day. Meanwhile, 16.4% of the GPS data from the global dataset had a suboptimal distance per day.

There was discrepancy between the data retrieved from questionnaire on day 14 and GPS data (optimal data, standardised time): participants could declare a visit in a specific

**Table 3. Missing GPS tracks and reasoning.**

| Missing GPS tracks | Rainy season | Dry season | All |
|---|---|---|---|
| Lost device | 7 | 8 | 15 |
| Damaged device | 54 | 23 | 77 |
| Participant withdrawal | 4 | 4 | 8 |
| Participant out of reach | 5 | 5 | 10 |
| Participant death | 1 | 0 | 1 |
| Total missing GPS tracks | 71 | 40 | 111 |
| Total expected GPS tracks | 320 | 400 | 720 |
| GPS data loss (%) | 22.2 | 10 | 15.4 |

environment, without any corresponding visit in their GPS data, and vice-versa (S6 Table). Overall, data discrepancy between GPS and questionnaire data varied from 26.9% (forest visits; over 197 participants, 53 had contradictive data between GPS and questionnaire) to 48.2% (fields visits, S7 Table).

## Discussion

Curated data provides useful insight about GPS tracking implementation to a rural Cambodian population, though results interpretation can be subject to numerous biases. Recall bias during questionnaires may have led to erroneous estimations of device removals. The data loggers may attribute time spent in a given land to a participant incorrectly, as accuracy depends on the land use surrounding the device, although the grid definition (100-metre cells) mitigates the effects from accuracy error (maximum = 17.2m) and precision error (maximum = 15.4m).

Though participants who declared device removal did not constitute a significant proportion of suboptimal data, over the whole dataset, most of the suboptimal data seems to have come from participants that did not wear their GPS data loggers during their daily activities and did not declare these removals during the questionnaire. This is suggested by the 129 answers that did not declare a visit to a specific environment while the corresponding GPS data did include such a visit (S6 Table).

The differences between suboptimal and optimal datasets suggest that most of the removed points were from stationary positions at night in built-up areas. Performing mobility analyses on the complete dataset, including the suboptimal data, would not be representative of the true population mobility, as time spent in built-up areas would be over-represented.

The proportion of suboptimal data from devices left at home, in addition to GPS malfunctions and hardware damage, highlights the importance of designing a study that considers this substantial data loss, with a methodology that accounts only for optimal data. The sample size should also account for this data depletion, in addition to participants leaving the study. For instance, Hast et al. [5] observed an important drop-out from their participants which limited the interpretation of their results.

The selection of optimal versus suboptimal data is also potentially subject to bias. Even though the study population habits were taken into account, the threshold of optimal data quality was defined artificially as a minimum of 48 hours logged over the week and 10 km traversed per day for each participant. We also assumed that the GPS tracks with a logged duration of less than 48 hours did not fully depict participants' movements, though it is still possible that a participant travelled only short distances that week. In addition, any participant who stayed at home most of the week, keeping his device on him at all times (the device would therefore go into hibernation mode), would be categorized as suboptimal even though his mobility would be adequately represented.

Participants' compliance to use GPS data loggers is often partial—participants are not always willing to carry the GPS data loggers at all times—as reported in previous studies using the devices for tracking populations at risk of malaria [7] and in other contexts [4,34,35]. In our study, we observed that most declared removals were minor and did not significantly affect the quality of the data. However, the data suggests that there were removals not declared during the questionnaire, enough to categorise 35.5% of retrieved data as suboptimal. As for participants unwilling to take the GPS devices with them for certain activities, we hypothesise that they would not acknowledge removing their loggers. However, the declared compliance of the participants was very high, with 90.3% reporting no removal of the GPS data logger during the follow-up. Vazquez-Prokopec et al. [35] also observed that participants can declare a very high

compliance (>80%) even though they did not carry the GPS loggers at all times. Consequently, it is essential to consider both technical issues and participant-induced biases, and to retain only the optimal data needed to estimate the mobility at individual level. These results highlight the need for a full data analysis and cleaning methodology with GIS when considering the use of GPS data loggers.

In the GMS, malaria exposure risk is associated with activities in and around forested areas [15,20,21]. The estimation of time spent in these environments at night is important for estimating individuals' exposure risk. The use of GPS tracking imported into a GIS environment can provide quantitative and precise data about mosquito exposure at a fine resolution [36]. Coupled with a follow-up to track recent [37] and active [38] malaria infections, this method can help pinpoint hotspot transmission areas that might require special attention to eradicate the disease.

Our study had two main limitations: participants' compliance and devices' suitability. Firstly, the representativeness of the GPS data was affected by participants withdrawing from the study or being out of reach, as well as by participants who did not take their devices with them for their daily mobility. Revising the enrolment strategy, or initiating education and awareness of local populations about malaria transmission would likely improve this aspect. Another option would be to further increase the sample size although this would come at a cost of time and funds. Secondly, the quality of the GPS devices was below expectations, despite manufacturers' specifications and even after some preliminary testing that validated the model and the settings. The I-gotU GT600 battery was not suited to log one week of GPS points according to the selected settings. The device robustness and water resistance were also not satisfactory. Different models should therefore be tested in order to find a device able to fulfil both the battery life requirements and sturdiness expectations under the given conditions. Higher-quality GPS tracking devices can be found on the market; however users have to balance the cost, sturdiness and size acceptable by the volunteers.

## Conclusion

The implementation of GPS follow-up for infectious disease research holds promising results. Our study showed what limitations need to be considered during study design and data analysis. Here, GPS trackers were decently accepted by the population. The accuracy and precision of GPS data loggers were concordant with the findings from other studies and corresponded to our requirements. After curating the data and accounting for suboptimal and missing tracks, it effectively described participants' daily movements. We insist on anticipating harsh climatic and working conditions, in addition to the necessity for thorough data pre-processing to remove GPS tracks from participants unwilling to bring their devices to their daily activities while declaring no removals during the questionnaire. Thus, GPS follow-ups with careful curation have the potential to generate detailed daily data at the individual level for rural populations. Such movement patterns can be used to refine epidemiological models and to adapt public health intervention strategies to focus on populations of interest.

## Supporting information

**S1 Table. Count of logged and analysed GPS points.**
(DOCX)

**S2 Table. GPS devices' accuracy controlled as the average error between individual recorded positions and true georeference given by a precise device.**
(DOCX)

**S3 Table. GPS devices' precision controlled as the average error between individual recorded positions and centroid of the units' recorded positions.**
(DOCX)

**S4 Table. Proportions of tracks belonging to insufficient, poor and optimal categories, allowing selecting optimal GPS tracks for analysis.**
(DOCX)

**S5 Table. Percentage of every land use, speed and time variables attributed to suboptimal and optimal datasets.** Every land use category was compared to the total time logged minus the time in this land use category using a Chi-squared test. According to Bonferonni correction, p-value was significant when $p < 0.05/6 = 0.0083$.
(DOCX)

**S6 Table. Count of data discrepancies between questionnaire and GPS datasets about participants visits in forest, plantations and fields, for all participants with a complete dataset for questionnaire at week 2 and GPS follow-up (N = 197).** Discordant data is highlighted in yellow.
(DOCX)

**S7 Table. Total discrepancies between GPS and questionnaire data for participants with a complete dataset (N = 197).**
(DOCX)

## Acknowledgments

The authors express their gratitude to the study participants, village heads, VMWs and officials for making this study possible. The authors thank Anthony Ruberto for proofreading the manuscript.

## Author Contributions

**Conceptualization:** Anaïs Pepey, Benoit Witkowski, Amélie Vantaux.

**Data curation:** Anaïs Pepey, Thomas Obadia, Marc Souris.

**Formal analysis:** Anaïs Pepey, Amélie Vantaux, Marc Souris.

**Funding acquisition:** Ivo Mueller, Benoit Witkowski.

**Investigation:** Anaïs Pepey, Saorin Kim, Amélie Vantaux, Marc Souris.

**Methodology:** Anaïs Pepey, Thomas Obadia, Amélie Vantaux, Marc Souris.

**Project administration:** Ivo Mueller, Benoit Witkowski, Amélie Vantaux.

**Resources:** Benoit Witkowski.

**Software:** Marc Souris.

**Supervision:** Ivo Mueller, Benoit Witkowski, Amélie Vantaux, Marc Souris.

**Validation:** Siv Sovannaroth, Ivo Mueller, Benoit Witkowski, Amélie Vantaux, Marc Souris.

**Writing – original draft:** Anaïs Pepey.

**Writing – review & editing:** Anaïs Pepey, Thomas Obadia, Benoit Witkowski, Amélie Vantaux, Marc Souris.

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
