## [Decision Letter · Decision Letter 0]

22 Nov 2021

PONE-D-21-26876Mobility evaluation by GPS tracking for epidemiological studies in a rural, low-income population in CambodiaPLOS ONE

Dear Dr. Pepey,

Thank you for submitting your manuscript to PLOS ONE. We have finally secured the reviews and recommendations from two reviewers. After careful consideration, we feel that it has merit but does not fully meet PLOS ONE’s publication criteria as it currently stands. In particular, the manuscript needs to properly connect the GPS data collected with malaria exposure and transmission. Therefore, we invite you to submit a revised version of the manuscript that addresses all points raised during the review process. It is important that responses to the review comments are addressed directly in the manuscript and not only in the response letter. Furthermore, the English language needs to be improved and streamlined, thus it is suggested that the manuscript goes through a thorough copy-editing by a native English speaker before submission of the revised manuscript.

We look forward to receiving your revised manuscript.

Kind regards,

Andrés Viña

Academic Editor

PLOS ONE

Journal Requirements:

2. We note that Figure 2 in your submission contain [map/satellite] images which may be copyrighted. All PLOS content is published under the Creative Commons Attribution License (CC BY 4.0), which means that the manuscript, images, and Supporting Information files will be freely available online, and any third party is permitted to access, download, copy, distribute, and use these materials in any way, even commercially, with proper attribution. For these reasons, we cannot publish previously copyrighted maps or satellite images created using proprietary data, such as Google software (Google Maps, Street View, and Earth). For more information, see our copyright guidelines: http://journals.plos.org/plosone/s/licenses-and-copyright.

Reviewers' comments:

Reviewer's Responses to Questions

**Comments to the Author**

1. Is the manuscript technically sound, and do the data support the conclusions?

Reviewer #1: Yes

Reviewer #2: No

2. Has the statistical analysis been performed appropriately and rigorously? 

Reviewer #1: Yes

Reviewer #2: No

3. Have the authors made all data underlying the findings in their manuscript fully available?

Reviewer #1: No

Reviewer #2: No

4. Is the manuscript presented in an intelligible fashion and written in standard English?

Reviewer #1: Yes

Reviewer #2: No

5. Review Comments to the Author

Reviewer #1: This manuscript presents the application of GPS data-loggers to track human mobility in rural Cambodia, focused on population impacted by malaria in the area.

1) The description of methods and technology, while not completely novel, provide useful information for future studies linking human mobility and malaria. There are several studies already presenting parameters for the GPS units used. This study provides a description of the barriers and opportunities of the use of the technology within the specific conditions of rural Cambodia. What the paper is missing, though, is a clear link between the data collected and the specific hypothesis regarding malaria exposure and transmission. I encourage the authors to expand their introduction to add more context to the link between mobility and malaria.

2) for instance, a specific age group was selected for study (13-60 yearolds). The rationale for their selection is missing. Provide evidence supporting this is a key group. Also, men are the target. Why?

3) Provide a justification for the collection of data every 30 sec. Seems a high frequency for mobility estimates, which come with a trade off in battery life.

4) Tables include a category 'other' but no explanation of what goes into it is provided.

5) What the paper is missing is a summary of the data showing any insight into their hypothesis. I strongly encourage authors to show some results of the breakdown of locations visited or whether there was any indication by age of different risk behavior. Anything that can show that the GPS units are collecting the data the authors are interested in gathering to test their overarching hypothesis. Also, this analysis will provide readers with an opportunity to appreciate the value of the data collected by GPS units. This is a major gap in the current draft.

6) Include a paragraph with limitations of the study.

Reviewer #2: This study proposes a method for processing GPS data to represent forest goers’ mobility patterns. Because the paper's title is "Mobility evaluation by GPS tracking for epidemiological studies in a rural, low-income population in Cambodia", I expected that this paper would present results of the exposure or risk assessments using GPS data for forest goers after they processed the GPS data, and then demonstrate how their mobility patterns pose a high risk for Malaria. However, their main body heavily focuses on their data collection protocol/procedure, how they processed and cleaned the GPS data, what the challenges were in the process (e.g., missing GPS tracks), and the data quality, and there was no assessment of individual-level exposure or risk for Malaria using the GPS data they cleaned.

The method that they proposed has some useful details about how to produce an optimal GPS data set, and those details may benefit the scientific communities that use GPS data in general, but I don’t see much contribution to epidemiology or public health research. I believe that contribution of a paper should be more than just the descriptions of data and data cleaning process. This paper also requires significant English editing (sometimes it was difficult to understand what the authors meant).

6. PLOS authors have the option to publish the peer review history of their article (what does this mean?). If published, this will include your full peer review and any attached files.

Reviewer #1: No

Reviewer #2: No

---

## [Author Response · Author response to Decision Letter 0]

3 Jan 2022

Response letter to PLOS ONE: PONE-D-21-26876

We thank the reviewers and the editor for their comments that helped to improve our manuscript. 

Reviewers' comments:

Reviewer #1: 

This manuscript presents the application of GPS data-loggers to track human mobility in rural Cambodia, focused on population impacted by malaria in the area.

1) The description of methods and technology, while not completely novel, provide useful information or future studies linking human mobility and malaria. There are several studies already presenting parameters for the GPS units used. This study provides a description of the barriers and opportunities of the use of the technology within the specific conditions of rural Cambodia. What the paper is missing, though, is a clear link between the data collected and the specific hypothesis regarding malaria exposure and transmission. I encourage the authors to expand their introduction to add more context to the link between mobility and malaria. 

We added a brief description of malaria dynamics in Cambodia and the importance of mobility as a risk factor (lines 60-70). 

2) for instance, a specific age group was selected for study (13-60 year olds). The rationale for their selection is missing. Provide evidence supporting this is a key group. Also, men are the target. Why?

Young and adult males represent the majority of the population at risk of malaria in the GMS (line 61). 

3) Provide a justification for the collection of data every 30 sec. Seems a high frequency for mobility estimates, which come with a trade off in battery life.

This interval was justified as very fine-scale mobility can be decisive in Cambodia (line 66-68). Moreover, initial tests allowed a week of recording with such settings (line 284-285).

4) Tables include a category 'other' but no explanation of what goes into it is provided.

Table 3 was updated and other is now indicating participant death category.

5) What the paper is missing is a summary of the data showing any insight into their hypothesis. I strongly encourage authors to show some results of the breakdown of locations visited or whether there was any indication by age of different risk behavior. Anything that can show that the GPS units are collecting the data the authors are interested in gathering to test their overarching hypothesis. Also, this analysis will provide readers with an opportunity to appreciate the value of the data collected by GPS units. This is a major gap in the current draft.

The aim of the manuscript was not to present an epidemiological study with risk factors analyses but rather the constraints to implement such work and a comparison of biases between GPS and questionnaires. We have now changed the title in order to avoid misleading the reader. 

6) Include a paragraph with limitations of the study.

We restructured the discussion and included a “limitations” paragraph (lines 278-290). 

Reviewer #2: 

This study proposes a method for processing GPS data to represent forest goers’ mobility patterns. Because the paper's title is "Mobility evaluation by GPS tracking for epidemiological studies in a rural, low-income population in Cambodia", I expected that this paper would present results of the exposure or risk assessments using GPS data for forest goers after they processed the GPS data, and then demonstrate how their mobility patterns pose a high risk for Malaria. However, their main body heavily focuses on their data collection protocol/procedure, how they processed and cleaned the GPS data, what the challenges were in the process (e.g., missing GPS tracks), and the data quality, and there was no assessment of individual-level exposure or risk for Malaria using the GPS data they cleaned.

The method that they proposed has some useful details about how to produce an optimal GPS data set, and those details may benefit the scientific communities that use GPS data in general, but I don’t see much contribution to epidemiology or public health research. I believe that contribution of a paper should be more than just the descriptions of data and data cleaning process. This paper also requires significant English editing (sometimes it was difficult to understand what the authors meant).

The method that we propose provides useful details about how to implement a GPS tracking and produce an optimal GPS data set, and those details may benefit the scientific communities that use GPS follow-ups, including epidemiology. We did not include spatial-temporal analyses and risk behaviour of participants regarding malaria exposure as the manuscript would have been too long and most importantly would have mixed different aspects thus blurring the take home messages. Consequently, we decided to separate the methodology from the risk factors analysis in order to convey the results and take-home messages more clearly. The manuscript has now been copy-edited by a native English speaker.

Journal Requirements:

Formatting was updated.

2. We note that Figure 2 in your submission contain [map/satellite] images which may be copyrighted. All PLOS content is published under the Creative Commons Attribution License (CC BY 4.0), which means that the manuscript, images, and Supporting Information files will be freely available online, and any third party is permitted to access, download, copy, distribute, and use these materials in any way, even commercially, with proper attribution. For these reasons, we cannot publish previously copyrighted maps or satellite images created using proprietary data, such as Google software (Google Maps, Street View, and Earth). For more information, see our copyright guidelines: http://journals.plos.org/plosone/s/

licenses-and-copyright.

The land use map was published in an open access journal in 2020, and appropriate citing was added.

To comply with both the journal requirements and the reviewer’s comments, all relevant anonymised datasets were published in the following public repository:

https://datadryad.org/stash/share/aUECweQYHVKmFzajwkLCacKaYBKh3j2wj5ufaRlaB8A

---

## [Decision Letter · Decision Letter 1]

22 Mar 2022

Mobility evaluation by GPS tracking in a rural, low-income population in Cambodia

PONE-D-21-26876R1

Dear Dr. Pepey,

We’re pleased to inform you that your manuscript has been judged scientifically suitable for publication and will be formally accepted for publication once it meets all outstanding technical requirements.

Kind regards,

Andrés Viña

Academic Editor

PLOS ONE

Additional Editor Comments (optional):

The authors have properly revised their manuscript based on the reviewers' comments and it is now ready to be accepted for publication in the pages of PLOS ONE.

Reviewers' comments:

Reviewer's Responses to Questions

**Comments to the Author**

1. If the authors have adequately addressed your comments raised in a previous round of review and you feel that this manuscript is now acceptable for publication, you may indicate that here to bypass the “Comments to the Author” section, enter your conflict of interest statement in the “Confidential to Editor” section, and submit your "Accept" recommendation.

Reviewer #1: All comments have been addressed

2. Is the manuscript technically sound, and do the data support the conclusions?

Reviewer #1: Yes

3. Has the statistical analysis been performed appropriately and rigorously? 

Reviewer #1: Yes

4. Have the authors made all data underlying the findings in their manuscript fully available?

Reviewer #1: Yes

5. Is the manuscript presented in an intelligible fashion and written in standard English?

Reviewer #1: Yes

6. Review Comments to the Author

Reviewer #1: The reviewers have addressed all my suggestions. I have no further comments about this submission. I like the new map with landuse information.

7. PLOS authors have the option to publish the peer review history of their article (what does this mean?). If published, this will include your full peer review and any attached files.

Reviewer #1: No

---

## [Editor Report · Acceptance letter]

4 May 2022

PONE-D-21-26876R1 

Mobility evaluation by GPS tracking in a rural, low-income population in Cambodia 

Dear Dr. Pepey:

I'm pleased to inform you that your manuscript has been deemed suitable for publication in PLOS ONE. Congratulations! Your manuscript is now with our production department. 

Kind regards, 

on behalf of

Dr. Andrés Viña 

Academic Editor

PLOS ONE